# Tackling Visual Control via Multi-View Exploration Maximization

## Abstract

We present MEM: **M**ulti-view **E**xploration **M**aximization for tackling complex visual control tasks. To the best of our knowledge, MEM is the first approach that combines multi-view representation learning and intrinsic reward-driven exploration in reinforcement learning (RL). More specifically, MEM first extracts the specific and shared information of multi-view observations to form high-quality features before performing RL on the learned features, enabling the agent to fully comprehend the environment and yield better actions. Furthermore, MEM transforms the multi-view features into intrinsic rewards based on entropy maximization to encourage exploration. As a result, MEM can significantly promote the sample-efficiency and generalization ability of the RL agent, facilitating solving real-world problems with high-dimensional observations and spare-reward space. We evaluate MEM on various tasks from DeepMind Control Suite and Procgen games. Extensive simulation results demonstrate that MEM can achieve superior performance and outperform the benchmarking schemes with simple architecture and higher efficiency.

## 1 Introduction

Achieving efficient and robust visual control towards real-world scenarios remains a long-standing challenge in deep reinforcement learning (RL). The critical problems can be summarized as (i) the partial observability of environments, (ii) learning effective representations of visual observations, and (iii) sparse or even missing reward feedback (Mnih et al., 2015; Lillicrap et al., 2016; Kalashnikov et al., 2018; Badia et al., 2020). To address the problem of partial observability, a simple approach is to construct the state by stacking consecutive frames (Mnih et al., 2015) or use the recurrent neural network to capture the environment dynamics (Hausknecht & Stone, 2015). However, most existing RL algorithms only consider observations from a single view of state space, which are limited by insufficient information. Multiple images captured from different viewpoints or times can provide more reference knowledge, making the task much easier. For instance, an autonomous vehicle will use multiple sensors to sense road conditions and take safe actions. Therefore, Li et al. (2019) propose the concept of multi-view RL and extend the partially observable Markov decision process (POMDP) to support multiple observation models. As a result, the agent can obtain more environment information and learn robust policies with higher generalization ability across domains.

However, it is always challenging to learn comprehensive low-dimensional representations from high-dimensional observations such as raw pixels. To address the problem, Hafner et al. (2019b) use an auto-encoder to learn the environment dynamics from raw pixels and select actions via fast online planning in the latent space. Yarats et al. (2021b) insert a deterministic auto-encoder to model-free RL algorithms and jointly trains the policy and latent state space, which is shown to be successful on tasks with noisy observations. (Stooke et al., 2021; Schwarzer et al., 2021) use contrastive learning to extract high-level features from raw pixels and subsequently perform off-policy control using the extracted features. In contrast, Laskin et al. (2020) propose to perform data augmentations (e.g., random translate and random amplitude scale) for RL on inputs, enabling simple RL algorithms to outperform complex methods.

Despite their good performance, using visual representation learning produces significant computational complexity, and imperfect representation learning may incur severe performance loss. Moreover, they cannot

Table 1: Comparing MEM with related work. RAD (Laskin et al., 2020), DrAC Raileanu et al. (2021), CURL (Srinivas et al., 2020), RE3 (Seo et al., 2021), DRIBO (Fan & Li, 2022). **Repr.**: the method involves special visual representation learning techniques. **Multi-view**: the method considers multi-view observations. **Exploration**: the method can encourage exploration. **Visual**: the method works well in visual RL.

| Method | Key insight | Repr. | Multi-view | Exploration | Visual |
|---|---|---|---|---|---|
| RAD | Data augmentation | ✓ | ✓ | ✗ | ✓ |
| DrAC | Data augmentation | ✗ | ✓ | ✗ | ✓ |
| CURL | Contrastive learning | ✓ | ✗ | ✗ | ✓ |
| RE3 | Intrinsic reward | ✗ | ✗ | ✓ | ✓ |
| DRIBO | Multi-view information bottleneck | ✓ | ✓ | ✗ | ✓ |
| MEM (ours) | Multi-view encoding and intrinsic reward | ✓ | ✓ | ✓ | ✓ |

handle environments with sparse reward space, and the learned policies are sensitive to visual distractions (Schmidhuber, 1991; Oudeyer & Kaplan, 2008; Oudeyer et al., 2007). To address the problem, recent approaches propose to leverage intrinsic rewards to improve the sample-efficiency and generalization ability of RL agents (Pathak et al., 2017; Burda et al., 2019b; Stadie et al., 2015). Burda et al. (2019a) demonstrate that RL agents can achieve surprising performance in various visual tasks using only intrinsic rewards. Seo et al. (2021) design a state entropy-based intrinsic reward module entitled RE3, which requires no representation learning and can be combined with arbitrary RL algorithms. Simulation results demonstrate that RE3 can significantly promote the sample-efficiency of RL agents both in continuous and discrete control tasks. Meanwhile, (Raileanu & Rocktäschel, 2020) utilize the difference between two consecutive states as the intrinsic reward, encouraging the agent to take actions that result in large state changes. This allows the agent to realize aggressive exploration and effectively adapt to the procedurally-generated environments.

In this paper, we deeply consider the three critical challenges above and propose a novel framework entitled **M**ulti-view **E**xploration **M**aximization (MEM) that exploits multi-view environment information to facilitate visual control tasks. Our contributions can be summarized as follows:

- Firstly, we introduce a novel multi-view representation learning method to learn high-quality features of multi-view observations to achieve efficient visual control. In particular, our multi-view encoder has simple architecture and is compatible with any number of viewpoints.

- Secondly, we combine multi-view representation learning and intrinsic reward-driven exploration, which computes intrinsic rewards using the learned multi-view features and significantly improves the sample-efficiency and generalization ability of RL algorithms. To the best of our knowledge, this is the first work that introduces the concept of multi-view exploration maximization. We also provide a detailed comparison between MEM and other representative methods in Table 1.

- Finally, we test MEM using multiple complex tasks of DeepMind Control Suite and Procgen games. In particular, we use two cameras to generate observations in the tasks of DeepMind Control Suite to simulate more realistic multi-view scenarios. Extensive simulation results demonstrate that MEM can achieve superior performance and outperform the benchmarking methods with higher efficiency and generalization ability.

## 2 Related Work

### 2.1 Multi-View Representation Learning

Multi-view representation learning aims to learn features of multi-view data to facilitate developing prediction models (Zhao et al., 2017). Most existing multi-view representation learning methods can be categorized

into three paradigms, namely the joint representation, alignment representation, and shared and specific representation, respectively (Jia et al., 2020). Joint representation integrates multiple views via concatenation (Tao et al., 2017; Srivastava & Salakhutdinov, 2012; Nie et al., 2017) while alignment representation maximizes the agreement among representations learned from different views (Chen et al., 2012; Frome et al., 2013; Jing et al., 2017). However, the former two methods suffer from the problem of information redundancy and the loss of complementary information, respectively. Shared and specific representation overcomes their shortcomings by disentangling the shared and specific information of different views and only aligning the shared part. In this paper, we follow the third paradigm and employ a simple encoder to learn comprehensive features from multi-view observations.

## 2.2 Representation Learning in RL

A common and effective workflow for visual control tasks is to learn representations from raw observations before training RL agents using the learned representations. Lee et al. (2020) integrate stochastic model learning and RL into a single method, achieving higher sample-efficiency and good asymptotic performance of model-free RL. (Hafner et al., 2019a; 2020) leverage variational inference to learn world models and solve long-horizon tasks by latent imagination. Mazoure et al. (2020) maximize concordance between representations using an auxiliary contrastive objective to increase predictive properties of the representations conditioned on actions. Schwarzer et al. (2021) train the agent to predict future states generated by a target encoder and learn temporally predictive and consistent representations of observations from different views, which significantly promotes the data efficiency of DRL agents. Yarats et al. (2020) demonstrate that data augmentation effectively promotes performance in visual control tasks and proposes an optimality invariant image transformation method for action-value function. In this paper, we consider multi-view observation space and use multi-view representation learning to extract low-dimensional features from raw observations. A closely related work to us is DRIBO proposed by (Fan & Li, 2022). DRIBO introduces a multi-view information bottleneck to capture task-relevant information from multi-view observations and produces robust policies to visual distractions that can be generalized to unseen environments Federici et al. (2019). However, DRIBO overemphasizes the shared information of different viewpoints, resulting in the leak of complementary information. Moreover, the experiments performed in (Fan & Li, 2022) only use a single camera and apply background replacement to generate multi-view observations, which may provide insufficient multi-view information.

## 2.3 Intrinsic Reward-Driven Exploration

Intrinsic rewards have been widely combined with standard RL algorithms to improve the exploration and generalization ability of RL agents (Ostrovski et al., 2017; Badia et al., 2020; Yu et al., 2020; Yuan et al., 2022a). Strehl & Littman (2008) proposes to use state visitation counts as exploration bonuses in tabular settings to encourage the agent to revisit infrequently-seen states. Bellemare et al. (2016) further extend such methods to high-dimensional state space. Kim et al. (2019) define the exploration bonus by applying a Jensen-Shannon divergence-based lower bound on mutual information across subsequent frames. Badia et al. (2020) combine episodic state novelty and life-long state novelty as exploration bonuses, which prevents the intrinsic rewards from decreasing with visits and provides sustainable exploration incentives. Yuan et al. (2022b) propose to maximize the Rényi entropy of state visitation distribution and transform it into particle-based intrinsic rewards. In this paper, we compute intrinsic rewards using multi-view features to realize multi-view exploration maximization, which enables the agent to explore the environment from multiple perspectives and obtain more comprehensive information.

# 3 Preliminaries

## 3.1 Multi-View Reinforcement Learning

In this paper, we study the visual control problem considering a multi-view MDP (Li et al., 2019), which can be defined as a tuple $\mathcal{M} = \langle \mathcal{S}, \mathcal{A}, P, \mathcal{O}_1, P_{\text{obs}}^1, \ldots, \mathcal{O}_N, P_{\text{obs}}^N, \check{r}, \gamma \rangle$. Here, $\mathcal{S}$ is the state space, $\mathcal{A}$ is the action space, $P : \mathcal{S} \times \mathcal{A} \times \mathcal{S} \to [0, 1])$ is the state-transition model, $\mathcal{O}_i$ is the observation space of the $i$-th

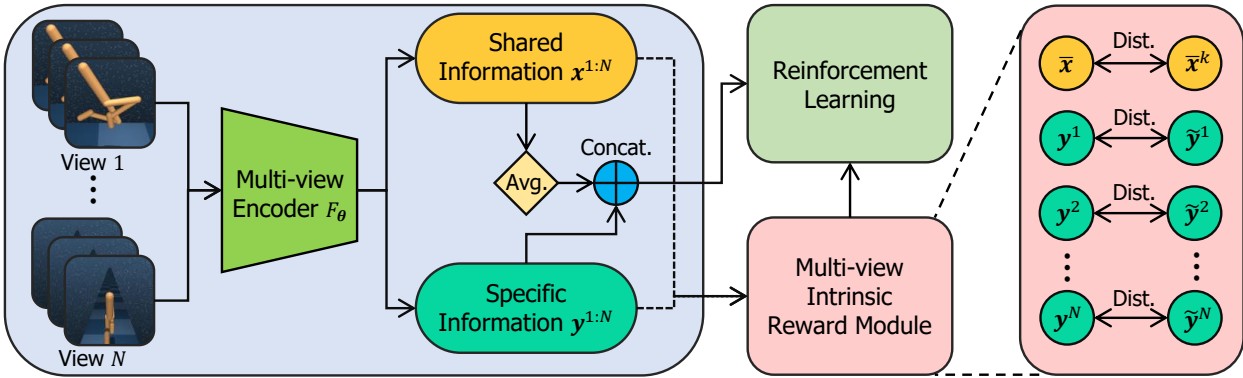

Figure 1: An overview of the MEM. Here, **avg.** represents the average operation, **concat.** represents the concatenate operation, and **dist.** represents the Euclidean distance.

viewpoint, $P_{\text{obs}}^i : \mathcal{S} \times \mathcal{O}_i \to [0, 1]$ is the observation model of the $i$-th viewpoint, $\check{r} : \mathcal{S} \times \mathcal{A} \to \mathbb{R}$ is the reward function, and $\gamma \in [0, 1)$ is a discount factor. In particular, we use $\check{r}$ to distinguish the extrinsic reward from the intrinsic reward $\hat{r}$ in the sequel. At each time step $t$, we construct the state $\boldsymbol{s}_t$ by encoding consecutive multi-view observations $\{\boldsymbol{o}_t^i, \boldsymbol{o}_{t-1}^i, \boldsymbol{o}_{t-2}^i, \dots\}_{i=1}^N$, where $\boldsymbol{o}_t^i \sim P_{\text{obs}}^i(\cdot|\boldsymbol{s}_t)$. Denoting by $\pi(\boldsymbol{a}_t|\boldsymbol{s}_t)$ the policy of the agent, the objective of RL is to learn a policy that maximizes the expected discounted return $R_t = \mathbb{E}_\pi \left[ \sum_{k=0}^\infty \gamma^k \check{r}_{t+k+1} \right]$ (Sutton & Barto, 2018).

### 3.2 Fast Entropy Estimation

In the following sections, we compute the multi-view intrinsic rewards based on Shannon entropy of state visitation distribution (Shannon, 1948). Since it is non-trivial to evaluate the entropy when handling complex environments with high-dimensional observations, a convenient estimator is introduced to realize efficient entropy estimation. Let $X_1, \dots, X_n$ denote the independent and identically distributed samples drawn from a distribution with density $p$, and the support of $p$ is a set $\mathcal{X} \subseteq \mathbb{R}^q$. The entropy of $p$ can be estimated using a particle-based estimator (Leonenko et al., 2008):

$$\hat{H}_n^k(p) = \frac{1}{n} \sum_{i=1}^n \log \frac{n \cdot \|X_i - \tilde{X}_i\|_2^q \cdot \hat{\pi}^{\frac{q}{2}}}{k \cdot \Gamma(\frac{q}{2} + 1)} + \log k - \Psi(k) \propto \frac{1}{n} \sum_{i=1}^n \log \|X_i - \tilde{X}_i\|_2, \tag{1}$$

where $\tilde{X}_i$ is the $k$-nearest neighbor of $X_i$ among $\{X_i\}_{i=1}^n$, $\Gamma(\cdot)$ is the Gamma function, $\hat{\pi} \approx 3.14159$, and $\Psi(\cdot)$ is the Digamma function.

## 4 MEM

In this section, we propose the MEM framework that performs visual control based on multi-view observations. As illustrated in Figure 1, MEM is composed of two key components, namely the multi-view encoder and the multi-view intrinsic reward module, respectively. At each time step, the multi-view encoder transforms the multi-view observations into shared and specific information, which is used to make an action. Meanwhile, the shared and specific information is sent to the multi-view intrinsic reward module to compute intrinsic rewards. Finally, the policy will be updated using the augmented rewards.

### 4.1 Multi-View Encoding

Figure 2 illustrates the architecture of our multi-view encoder, which has an encoding network and discriminator. Let $F_{\boldsymbol{\theta}}$ denote the encoding network with parameters $\boldsymbol{\theta}$ that has two branches to extract the inter-view shared and intra-view specific information of the observation, respectively. For the observation $\boldsymbol{o}^i$ from the $i$-th viewpoint, we have

$$(\boldsymbol{x}^i, \boldsymbol{y}^i) = F_{\boldsymbol{\theta}}(\boldsymbol{o}^i), \tag{2}$$

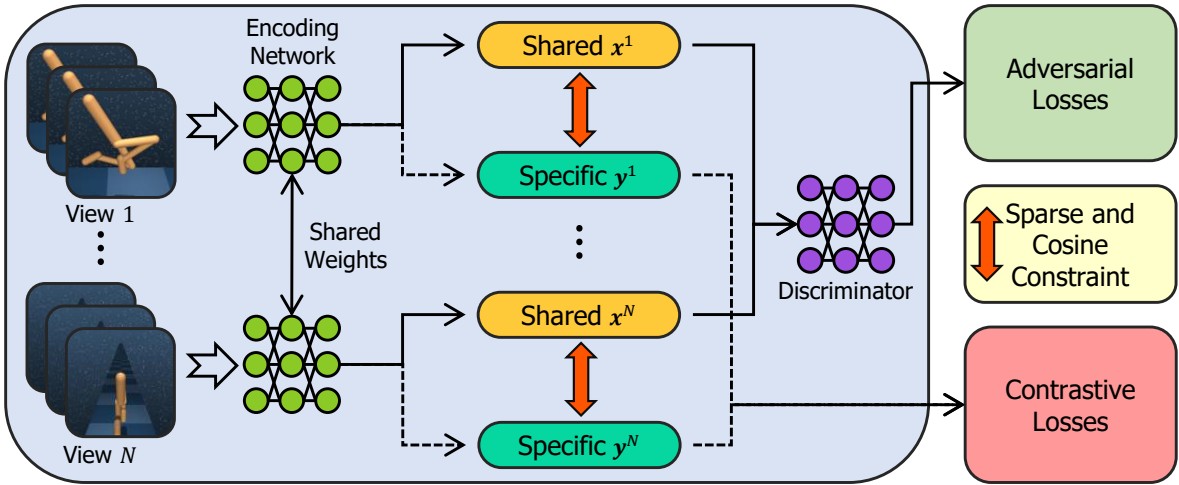

Figure 2: The architecture of the multi-view encoder.

where $\boldsymbol{x}^i$ is the inter-view shared information and $\boldsymbol{y}^i$ is the intra-view specific information. To separate the shared and specific information, we propose the following mixed constraints:

$$L_{\text{diff}} = \max\left\{\frac{<\boldsymbol{x}^i, \boldsymbol{y}^i>}{\|\boldsymbol{x}^i\|_2 \cdot \|\boldsymbol{y}^i\|_2}, 0\right\} + \|\boldsymbol{x}^i\|_1 + \|\boldsymbol{y}^i\|_1. \tag{3}$$

The first term of Eq. (3) is the cosine similarity of $\boldsymbol{x}^i$ and $\boldsymbol{y}^i$ to disentangle the two kinds of information. In particular, Liu et al. (2019) found that sparse representations can contribute to control tasks by providing locality and avoiding catastrophic interference. Therefore, the $\ell_1$-norm of $\boldsymbol{x}^i$ and $\boldsymbol{y}^i$ are leveraged to increase the sparsity of the learned features to improve the generalization ability.

To guarantee the discriminability of the specific information from multiple viewpoints, we follow the insight of the Siamese network that maximizes the distance between samples from different classes (Chopra et al., 2005). For a mini-batch, we define the contrastive loss as follows (Jia et al., 2020):

$$L_{\text{con}} = \frac{1}{2M} \sum_{j=1}^{M} \left[\|\boldsymbol{y}_j - \boldsymbol{\mu}_{\text{same}}\|_2^2 + \max^2(\text{Margin} - \|\boldsymbol{y}_j - \boldsymbol{\mu}_{\text{diff}}\|_2, 0)\right], \tag{4}$$

where $M$ is the batch size, $\boldsymbol{\mu}_{\text{same}}$ is the average of samples in this mini-batch from the same viewpoint with $\boldsymbol{y}_i$, and $\boldsymbol{\mu}_{\text{diff}}$ is the average of samples in this mini-batch from the different viewpoint with $\boldsymbol{y}_i$.

To extract the shared information, we follow (Jia et al., 2020) who design the similarity constraint using a generative adversarial pattern Goodfellow et al. (2014). More specifically, the encoding network is regarded as the generator, and the shared information is considered as the generated results. Meanwhile, a classification network is leveraged to serve as the discriminator. Therefore, the discriminator aims to judge the viewpoint of each shared information while the generator aims to fool the generator. Denoting by $D_{\phi}$ the discriminator represented by a neural network with parameters $\phi$, the loss function is

$$L_{\text{adv}} = \min_{\boldsymbol{\theta}} \max_{\boldsymbol{\phi}} \sum_{j=1}^{M} \sum_{i=1}^{N} l_j^i \cdot \log P(\boldsymbol{x}_j^i), \tag{5}$$

where $P(\boldsymbol{x}_j^i) = D_{\psi}(\boldsymbol{x}_j^i)$ is the probability that $\boldsymbol{x}_j^i$ is generated from the $i$-th viewpoint. Then the generator and discriminator will be trained until the discriminator cannot distinguish the differences between shared information of different viewpoints. Finally, the total loss of the multi-view encoder is

$$L_{\text{total}} = \lambda_1 \cdot L_{\text{diff}} + \lambda_2 \cdot L_{\text{con}} + \lambda_3 \cdot L_{\text{adv}}, \tag{6}$$

where $\lambda_1, \lambda_2, \lambda_3$ are the weighting coefficients.

Equipped with the shared and specific information, we define the state of timestep $t$ as

$$s_t = \text{Concatenate}(\boldsymbol{y}_t^1, \ldots, \boldsymbol{y}_t^N, \bar{\boldsymbol{x}}_t), \tag{7}$$

where $\bar{\boldsymbol{x}}_t = \frac{1}{N} \sum_{i=1}^N \boldsymbol{x}_t^i$ is the average of the shared inforamtion of $N$ viewpoints. Then the learned $s$ is sent to the agent to make actions.

### 4.2 Multi-View Intrinsic Reward

Next, we transform the learned features into intrinsic rewards to encourage exploration and promote sample-efficiency of the RL agent. Inspired by the work of Seo et al. (2021) and Yuan et al. (2022b), we propose to maximize the following entropy:

$$H(d) = -\mathbb{E}_{\boldsymbol{o}^i \sim d(\boldsymbol{o}^i)}[\log d(\boldsymbol{o}^i)], \tag{8}$$

where $d(\boldsymbol{o}^i)$ is the observation visitation distribution of the $i$-th viewpoint induced by policy $\pi$. Given multi-view observations of $T$ steps $\{\boldsymbol{o}_0^1, \boldsymbol{o}_1^1, \ldots, \boldsymbol{o}_T^1, \boldsymbol{o}_0^N, \boldsymbol{o}_1^N, \ldots, \boldsymbol{o}_T^N\}$, using Eq. (1), we define the multi-view intrinsic reward of the time step $t$ as

$$\hat{r}_t = \frac{1}{N+1} \left[ \left( \sum_{i=1}^N \log(\|\boldsymbol{y}_t^i - \tilde{\boldsymbol{y}}_t^i\|_2 + 1) \right) + \log(\|\bar{\boldsymbol{x}}_t - \bar{\boldsymbol{x}}_t^k\|_2 + 1) \right], \tag{9}$$

where $\tilde{\boldsymbol{y}}_t^i$ is the $k$-nearest neighbor of $\boldsymbol{y}_t^i$ among $\{\boldsymbol{y}_t^i\}_{t=0}^T$ and $\bar{\boldsymbol{x}}_t^k$ is the $k$-nearest neighbor of $\bar{\boldsymbol{x}}_t$.

We highlight the advantages of the proposed intrinsic reward. Firstly, $\hat{r}_t$ measures the distance between observations in the representation space. It encourages the agent to visit as many distinct parts of the environment as possible. Similar to RIDE of (Raileanu & Rocktäschel, 2020), $\hat{r}_t$ can also lead the agent to take actions that result in large state changes, which can facilitate solving procedurally-generated environments. Moreover, $\hat{r}_t$ evaluates the visitation entropy of multiple observation spaces and reflects the global exploration extent more comprehensively. Finally, the generation of $\hat{r}_t$ requires no memory model or database, which will not vanish as the training goes on and can provide sustainable exploration incentives.

### 4.3 Training Objective

Equipped with the intrinsic reward, the total reward of each transition $(\boldsymbol{s}_t, \boldsymbol{a}_t, \boldsymbol{s}_{t+1})$ is computed as

$$r_t^{\text{total}} = \check{r}(\boldsymbol{s}_t, \boldsymbol{a}_t) + \beta_t \cdot \hat{r}(\boldsymbol{s}_t), \tag{10}$$

where $\beta_t = \beta_0(1-\kappa)^t \geq 0$ is a weighting coefficient that controls the exploration preference, and $\kappa$ is a decay rate. In particular, this intrinsic reward can be leveraged to perform unsupervised pre-training without extrinsic rewards $\check{r}$. Then the pre-trained policy can be employed in the downstream task adaptation with extrinsic rewards. Letting $\pi_{\boldsymbol{\varphi}}$ denote the policy represented by a neural network with parameters $\boldsymbol{\varphi}$, the training objective of MEM is to maximize the expected discounted return $\mathbb{E}_{\pi_{\boldsymbol{\varphi}}} \left[ \sum_{k=0}^T \gamma^k r_{t+k+1}^{\text{total}} \right]$. Finally, the detailed workflows of MEM with off-policy RL and on-policy RL are summarized in Algorithm 1 and Appendix C, respectively.

## 5 Experiments

In this section, we designed the experiments to answer the following questions:

- Does the learned multi-view information contribute to achieving higher performance in visual control tasks? (See Figures 3 & 6 & 7)

- Can MEM outperform other schemes that involve multi-view representation learning and other representation learning techniques, such as contrastive learning? (See Figure 4 & 7)

---

**Algorithm 1:** MEM with Off-policy RL

---

Initialize encoding network $F_{\boldsymbol{\theta}}$ and discriminator $D_{\boldsymbol{\phi}}$;

Initialize policy network $\pi_{\boldsymbol{\varphi}}$, maximum environment steps $t_{\max}$, coefficient $\beta_0$, decay rate $\kappa$, and replay buffer $\mathcal{B} \leftarrow \emptyset$;

**for** step $t = 1, \ldots, t_{\max}$ **do**

    Get multi-view observation $\{\boldsymbol{o}_t^1, \ldots, \boldsymbol{o}_t^N\}$;

    **for** $i = 1, \ldots, N$ **do**

        $\boldsymbol{x}_t^i, \boldsymbol{y}_t^i = F_{\boldsymbol{\theta}}(\boldsymbol{o}_t^i)$;

    **end**

    Get state $\boldsymbol{s}_t = \text{Concatenate}(\boldsymbol{y}_t^1, \ldots, \boldsymbol{y}_t^N, \bar{\boldsymbol{x}}_t)$;

    Sample an action $\boldsymbol{a}_t \sim \pi_{\boldsymbol{\varphi}}(\cdot|\boldsymbol{s}_t)$;

    $\mathcal{B} \leftarrow \mathcal{B} \cup \{\boldsymbol{o}_t^{1:N}, \boldsymbol{a}_t, \check{r}_t, \boldsymbol{o}_{t+1}^{1:N}\}$;

    Sample a random mini-batch $\{\boldsymbol{o}_j^{1:N}, \boldsymbol{a}_j, \check{r}_j, \boldsymbol{o}_{j+1}^{1:N}\}_{j=1}^B \sim \mathcal{B}$;

    Get representations $\{\boldsymbol{x}_j^{1:N}, \boldsymbol{y}_j^{1:N}\}_{j=1}^B$ ;

    **for** $j = 1, \ldots, B$ **do**

        Compute the intrinsic reward $\hat{r}_j$ using Eq. (9);

        Update $\beta_t = \beta_0(1 - \kappa)^t$;

        Let $r_j^{\text{total}} = \check{r}_j + \beta_t \cdot \hat{r}_j$;

    **end**

    Update the policy network with $\{\boldsymbol{o}_j^{1:N}, \boldsymbol{a}_j, r_j^{\text{total}}, \boldsymbol{o}_{j+1}^{1:N}\}_{j=1}^B$ using any off-policy RL algorithms;

    Update $\boldsymbol{\theta}, \boldsymbol{\phi}$ to minimize $L_{\text{total}}$ in Eq. (6).

**end**

---

- How does MEM compare to other intrinsic reward-driven methods? (See Figures 4 & 6 & 7 & Table 3)

- Can MEM achieve remarkable performance in robotic manipulation tasks with sparse-reward space? (See Figure 6)

- How about the generalization ability of MEM in procedurally-generated environments? (See Table 2 & 3)

## 5.1 DeepMind Control Suite

### 5.1.1 Setup

We first tested MEM on six complex visual control tasks of DeepMind Control Suite, namely the *Cheetah Run*, *Finger Turn Hard*, *Hopper Hop*, *Quadruped Run*, *Reacher Hard*, and *Walker Run*, respectively (Tassa et al., 2018). To evaluate the sample-efficiency of MEM, two representative model-free RL algorithms Soft Actor-Critic (SAC) (Haarnoja et al., 2018) and Data Regularized Q-v2 (DrQ-v2) (Yarats et al., 2021a), were selected to serve as the baselines. For comparison with schemes that involve multi-view representation learning, we selected DRIBO of (Fan & Li, 2022) as the benchmarking method, which introduces a multi-view information bottleneck to maximize the mutual information between sequences of observations and sequences of representations. For comparison with schemes that involve contrastive representation learning, we selected CURL of (Srinivas et al., 2020), which maximizes the similarity between different augmentations of the same observation. For comparison with other exploration methods, we selected RE3 of (Seo et al., 2021) that maximizes the Shannon entropy of state visitation distribution using a random encoder and considered the combination of RE3 and DrQ-v2. The following results were obtained by setting $k = 3, \beta_0 = 0.05$ and $\kappa = 0.000025$, and more details are provided in Appendix A.

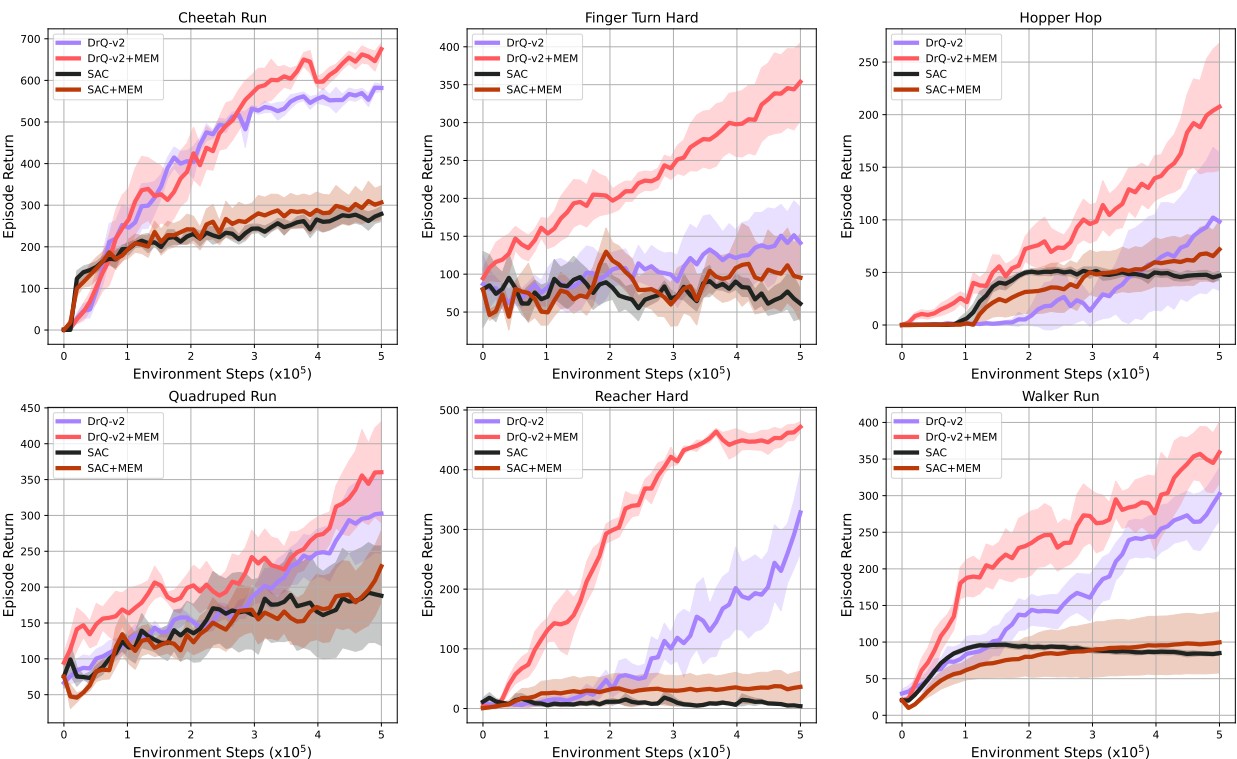

Figure 3: Performance on six complex control tasks from DeepMind Control Suite over 10 random seeds. MEM significantly promotes the sample efficiency of DrQ-v2. The solid line and shaded regions represent the mean and standard deviation, respectively.

### 5.1.2 Results

Figure 3 illustrates the comparison of the average episode return of six complex control tasks. Note that here SAC and DrQ-v2 only used **single-view** observations to perform RL. It is obvious that MEM significantly improved the sample-efficiency of DrQ-v2 on various tasks. In *Finger Turn Hard* task, DrQ-V2+MEM achieved an average episode return of 350.0, producing a big performance gain when compared to the vanilla DrQ-v2 agent. The multi-view observations allow the agent to observe the robot posture from multiple viewpoints, providing more straightforward feedback on the taken actions. As a result, the agent can adapt to the environment faster and achieve a higher convergence rate, especially for the *Hopper Hop* task and *Reacher Hard* task. In contrast, the SAC agent achieved low score in most tasks due to its performance limitation, and MEM only slightly promoted its sample-efficiency. Figure 3 proves that the multi-view observations indeed provide more comprehensive information about the environment, which helps the agent to fully comprehend the environment and make better decisions.

Next, we compared MEM with other methods that consider representation learning and intrinsic reward-driven exploration. As shown in Figure 4, DrQ-v2+MEM achieved the highest policy performance on all six tasks. Meanwhile, DrQ-v2+RE3 also performed impressive performance on various tasks, which also significantly promoted the sample-efficiency of the DrQ-v2 agent. In particular, DrQ-v2+MEM achieved an average episode return of 220.0 in *Hopper Hop* task, while DRIBO and CURL failed to solve the task. We also observed that CURL and DRIBO had lower convergence rates on various tasks, which may be caused by complex representation learning. Even if the design of representation learning is more sophisticated, the information extracted from single-view observations is still limited. In contrast, multi-view observations naturally contain more abundant information, and appropriate representation learning techniques will produce more helpful guidance for the agent.

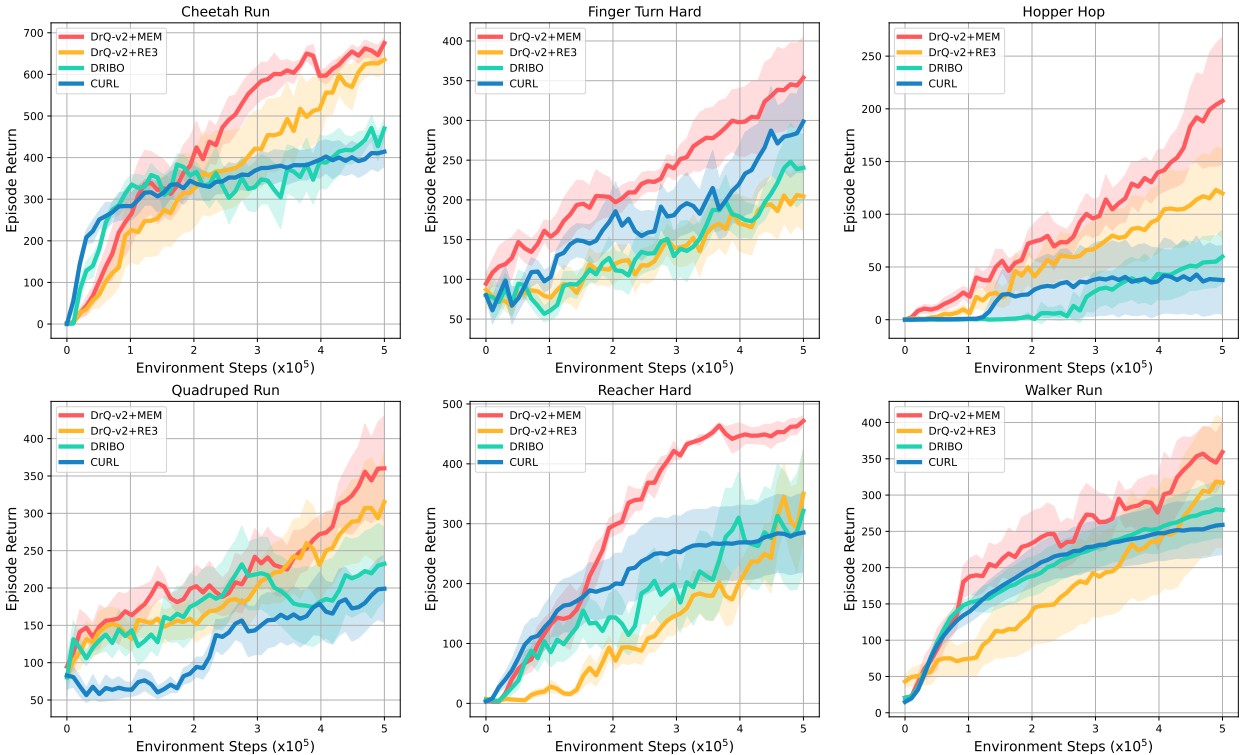

Figure 4: Performance on six complex control tasks from DeepMind Control Suite over 10 random seeds. MEM outperforms the benchmarking schemes in all the tasks. The solid line and shaded regions represent the mean and standard deviation, respectively.

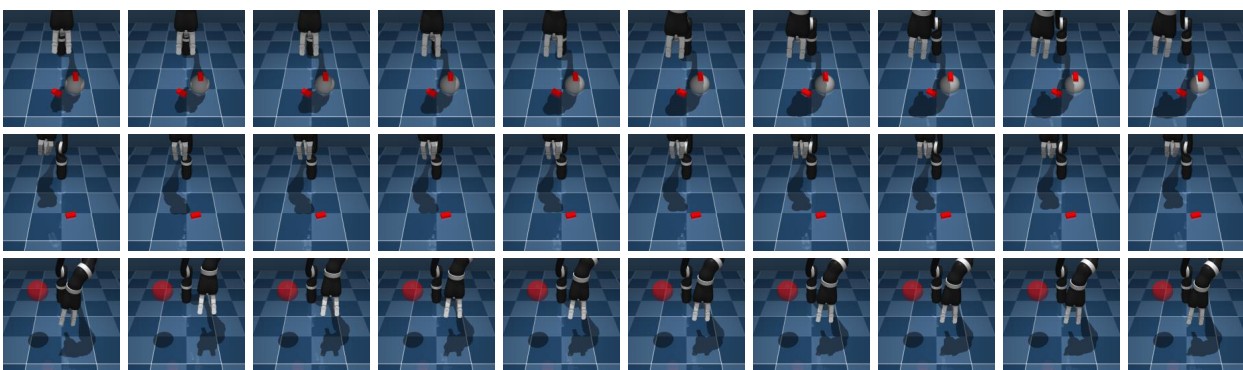

Figure 5: Rendered image observations for three robotic manipulation tasks: *Place Brick*, *Reach Duplo*, and *Reach Site* (from top to bottom).

To further investigate the increments of multi-view observations, we tested MEM on three robotic manipulation tasks with sparse rewards, which can evaluate its adaptability to real-world scenarios. Figure 5 illustrates the rendered image examples of these tasks. Figure 6 illustrates the performance comparison between MEM, RE3, and vanilla DrQ-v2 agents. MEM achieved the highest performance in all three tasks, especially in the *Reach Duplo* task. On the one hand, MEM provides high-quality intrinsic rewards to the agent to overcome the sparse reward space and promote sample-efficiency. On the other hand, multi-view observations provide more spatial and location information, enabling the agent to achieve more accurate positioning and movement. Therefore, MEM can effectively solve the manipulation tasks with fewer training

steps. Meanwhile, the combination of DrQ-v2 and RE3 also obtained remarkable performance on these tasks, which demonstrates the high effectiveness and efficiency of the intrinsic reward-driven exploration.

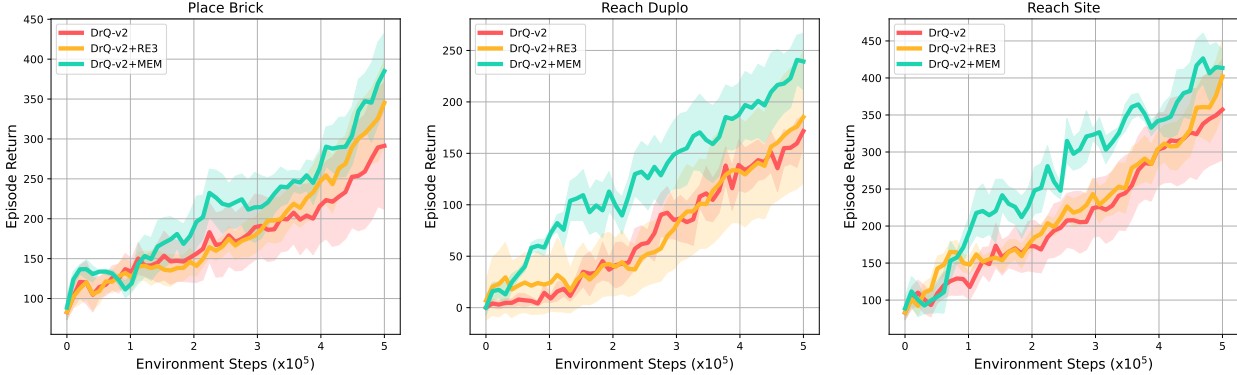

Figure 6: Performance on three manipulation tasks with sparse rewards over 10 random seeds. MEM can still achieve remarkable performance by providing high-quality intrinsic rewards. The solid line and shaded regions represent the mean and standard deviation, respectively.

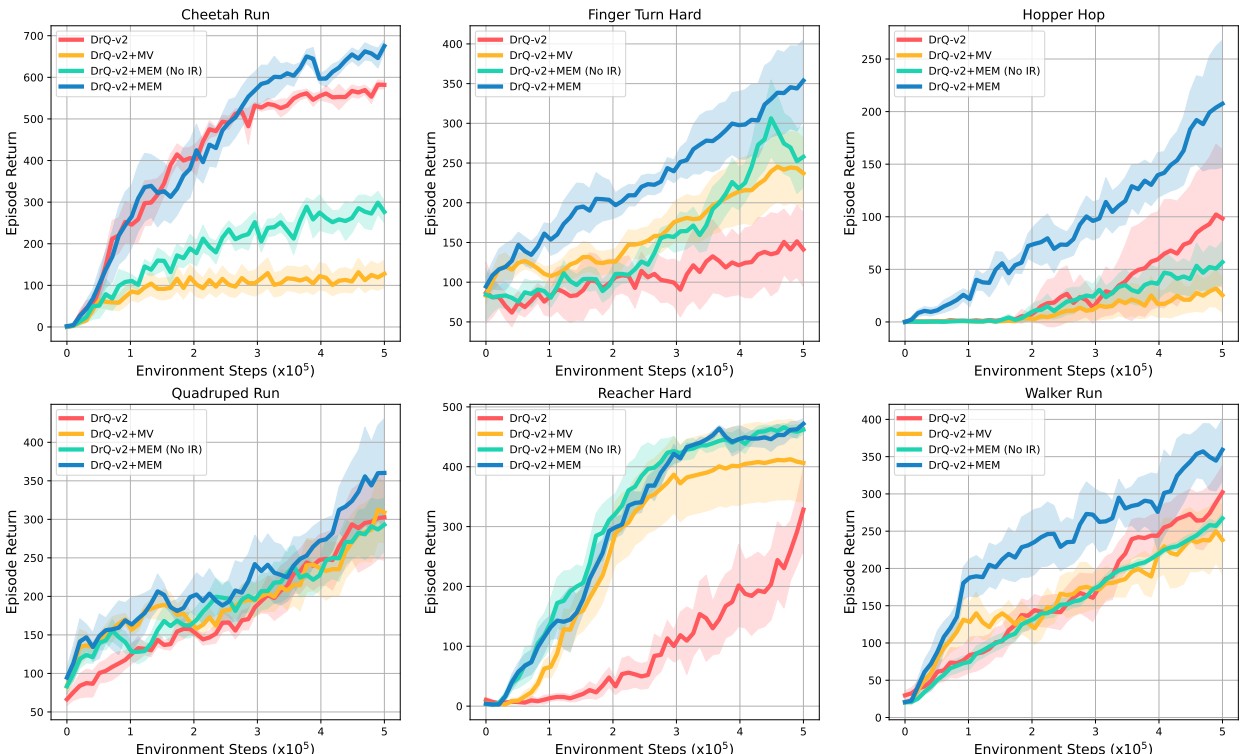

Figure 7: Performance on six complex control tasks from DeepMind Control Suite over 10 random seeds. The intrinsic reward module is shut down in DrQ-v2+MEM (No IR). The vanilla DrQ-v2 agent cannot handle multi-view observations. The solid line and shaded regions represent the mean and standard deviation, respectively.

## 5.2 Ablations

Since MEM benefits from two components: multi-view representation learning and intrinsic reward-driven exploration, we performed a number of ablations to emphasize the importance of each component used by

our method. First, DrQ-v2+MV was an ablation that learns directly from multi-view observations without using other representation learning techniques. This ablation helps disentangle the effect of using multi-view observations from the impact of using multi-view representation learning. Figure 7 illustrates the performance comparison between DrQ-v2, DrQ-v2+MV, DrQ-v2+MEM (No IR), and DrQ-v2+MEM on six complex control tasks, where the intrinsic reward module is shut down in DrQ-v2+MEM (No IR). The introduction of multi-view observations significantly improved the performance in *Finger Turn Hard* and *Reacher Hard*. However, DrQ-v2+MV failed to learn in *Cheetah Run* and *Hopper Hop*. In contrast, DrQ-v2+MEM (No IR) outperformed DrQ-v2+MV in all the tasks, which indicates that our multi-view encoding network can extract low-dimensional features from the observations effectively.

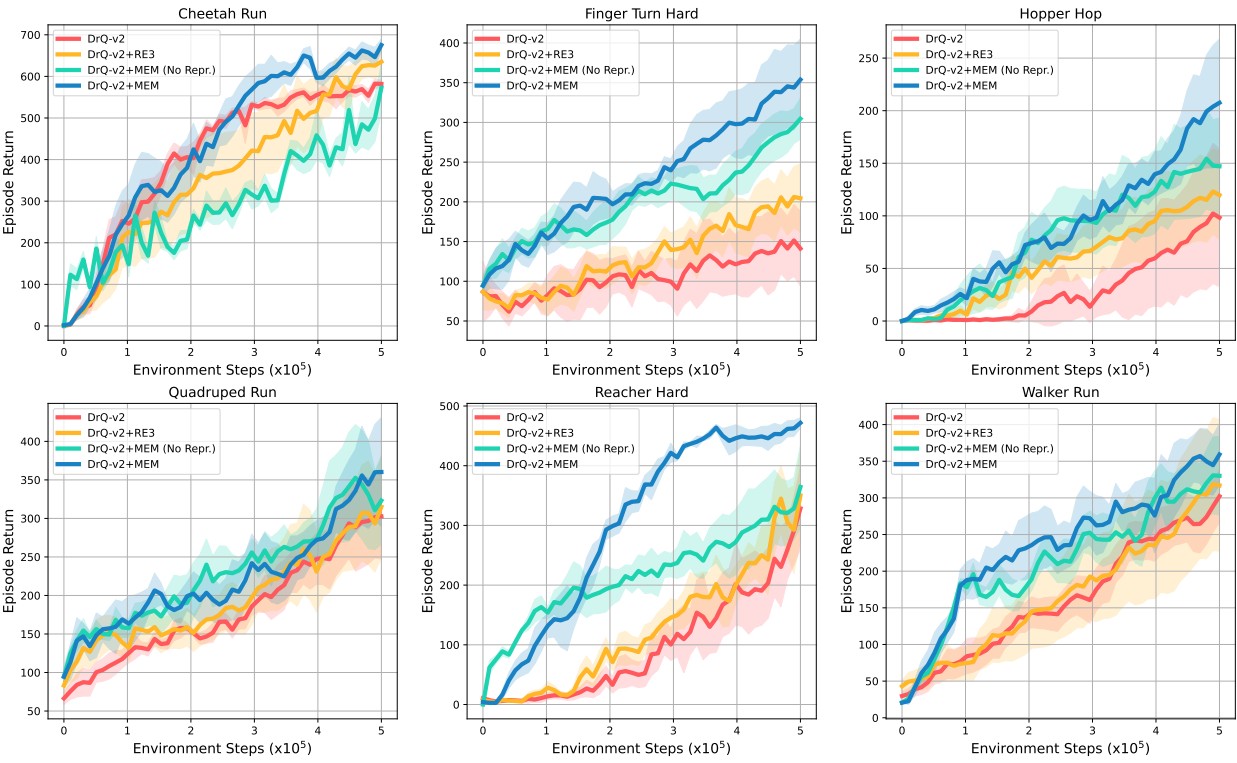

Figure 8: Performance on six complex control tasks from DeepMind Control Suite over 10 random seeds. The multi-view representation learning module is shut down in DrQ-v2+MEM (No Repr.). The intrinsic rewards derived by multi-view features produce significant sample-efficiency gain as compared to RE3. The solid line and shaded regions represent the mean and standard deviation, respectively.

The second ablation is DrQ-v2+MEM (No Repr.), in which the representation learning module is shut down. This ablation helps disentangle the effect of using multi-view representation learning from the impact of using multi-view intrinsic rewards. Moreover, it compares the increments produced by multi-view and single-view intrinsic rewards (RE3). Figure 7 illustrates the performance comparison between DrQ-v2, DrQ-v2+RE3, DrQ-v2+MEM (No Repr.), and DrQ-v2+MEM on six complex control tasks. The multi-view intrinsic rewards achieved higher performance than RE3 in five tasks, especially in *Finger Turn Hard* task. Considering the first ablation experiments, it is natural to find that the multi-view intrinsic rewards produce increments of MEM in most tasks, and multi-view features make the reward generation more accurate.

These results are consistent with our claim that (i) multi-view observations can effectively promote the agent performance under the supervision of multi-view representation learning, and (ii) evaluating intrinsic rewards with multiple observation spaces can enable the agent to explore the environment fully, and promote the sample efficiency significantly.

### 5.3 Procgen Games

Table 2: Performance on train levels after training 25M environment steps. The mean and standard deviation are computed over 10 random seeds.

| Game | PPO | UCB-DrAC | PPO+RE3 | PPO+RIDE | PPO+MEM |
|---|---|---|---|---|---|
| BigFish | 9.2±2.7 | 12.8±1.8 | 8.5±1.3 | 9.6±2.2 | **19.1±1.2** |
| BossFight | 8.0±0.4 | 8.1±0.4 | 9.6±0.4 | 9.4±0.2 | **9.7±0.4** |
| CaveFlyer | **6.7±0.6** | 5.8±0.9 | 4.9±0.6 | 5.7±0.3 | 6.0±0.3 |
| Chaser | 4.1±0.3 | **7.0±0.6** | 4.1±1.1 | 5.6±0.0 | 5.4±0.4 |
| Climber | 6.9±1.0 | 8.6±0.6 | 9.6±0.4 | 9.3±0.2 | **9.8±0.4** |
| CoinRun | 9.4±0.3 | 9.4±0.2 | **10.0±0.7** | **10.0±0.5** | **10.0±0.4** |
| Dodgeball | 5.3±2.3 | **7.3±0.8** | 3.2±0.2 | 4.4±0.3 | 5.4±0.4 |
| FruitBot | 28.8±0.6 | 29.3±0.5 | 28.9±1.5 | 29.5±1.8 | **30.3±0.3** |
| Heist | **7.1±0.5** | 6.2±0.6 | 4.7±0.3 | 5.2±0.3 | 5.7±0.4 |
| Jumper | 8.3±0.2 | 8.2±0.1 | 7.6±0.3 | 8.4±0.7 | **8.9±0.0** |
| Leaper | **5.5±0.4** | 5.0±0.9 | 4.2±0.3 | 4.3±0.3 | 4.2±0.4 |
| Maze | **9.1±0.2** | 8.5±0.3 | 5.9±0.3 | 5.9±0.7 | 6.6±0.4 |
| Miner | 11.7±0.5 | 12.0±0.3 | 10.6±0.1 | 12.2±1.7 | **12.4±0.2** |
| Ninja | 7.3±0.3 | 8.0±0.4 | 9.3±0.7 | **9.6±0.9** | 9.5±0.5 |
| Plunder | 6.1±0.8 | 10.2±1.76 | 10.3±0.8 | 11.2±0.6 | **13.2±1.3** |
| StarPilot | 29.0±1.1 | 33.1±1.3 | 33.0±1.3 | 33.5±2.1 | **37.1±0.6** |

Table 3: Performance on test levels after training 25M environment steps. The mean and standard deviation are computed over 10 random seeds.

| Game | PPO | UCB-DrAC | PPO+RE3 | PPO+RIDE | PPO+MEM |
|---|---|---|---|---|---|
| BigFish | 3.7±1.3 | 9.2±2.0 | 6.5±1.3 | 8.5±3.5 | **18.1±2.6** |
| BossFight | 7.4±0.4 | 7.8±0.6 | 9.2±0.2 | 8.8±0.8 | **9.9±0.4** |
| CaveFlyer | **5.1±0.4** | 5.0±0.8 | 4.7±0.6 | 4.7±0.7 | 4.8±0.4 |
| Chaser | 3.5±0.9 | 6.3±0.6 | 6.2±0.7 | 5.2±0.0 | 6.2±0.4 |
| Climber | 5.6±0.5 | 6.3±0.6 | **7.5±0.3** | 6.5±0.3 | **7.5±0.1** |
| CoinRun | 8.6±0.2 | 8.6±0.2 | 9.2±0.5 | 9.3±0.5 | **9.5±0.3** |
| Dodgeball | 1.6±0.1 | **4.2±0.9** | 2.7±0.1 | 2.7±0.1 | 3.4±1.0 |
| FruitBot | 26.2±1.2 | 27.6±0.4 | 27.9±1.5 | 28.6±0.6 | **30.0±0.4** |
| Heist | 2.5±0.6 | 3.5±0.4 | 3.4±0.0 | 3.4±0.3 | **3.7±0.2** |
| Jumper | 5.9±0.2 | 6.2±0.3 | 6.0±0.5 | 6.3±0.4 | **6.7±0.2** |
| Leaper | **4.9±2.2** | 4.8±0.9 | 3.6±0.3 | 3.6±0.7 | 4.2±0.7 |
| Maze | 5.5±0.3 | **6.3±0.1** | 5.7±1.0 | 5.4±0.3 | 5.9±0.5 |
| Miner | 8.4±0.7 | 9.2±0.6 | 5.8±1.3 | 6.7±0.0 | **9.6±0.5** |
| Ninja | 5.9±0.2 | **6.6±0.4** | 5.9±0.6 | 6.1±1.5 | 5.5±0.3 |
| Plunder | 5.2±0.6 | 8.3±1.1 | 11.2±0.1 | 12.1±0.3 | **12.5±0.9** |
| StarPilot | 24.9±1.0 | 30.0±1.3 | 32.7±2.0 | **34.1±2.4** | 32.5±2.7 |

### 5.3.1 Setup

Next, we tested MEM on nine Procgen games with graphic observations and discrete action space (Cobbe et al., 2020). Since Procgen games have procedurally-generated environments, it provides a direct measure to evaluate the generalization ability of an RL agent. We selected Proximal Policy Optimization (PPO) as the baseline (Schulman et al., 2017), and three approaches were selected to serve as the benchmarking methods, namely RIDE, RE3, and UCB-DrAC, respectively (Raileanu et al., 2021; Seo et al., 2021; Raileanu & Rocktäschel, 2020). RIDE uses the difference between consecutive states as intrinsic rewards, motivating

the agent to take actions that result in significant state changes. In contrast, UCB-DrAC tackles visual control tasks via data augmentation and uses upper confidence bound algorithm to automatically select an effective transformation for a given task. The following results were obtained by setting $k = 5, \beta_0 = 0.1$ and $\kappa = 0.00001$, and more details are provided in Appendix B.

### 5.3.2 Results

Table 2 and Table 3 illustrates the performance comparison between AIRS and benchmarking schemes on the full Procgen benchmark. For the train levels, the combination of PPO and MEM achieves a higher average episode return as compared to the vanilla PPO agent in most games. Meanwhile, MEM outperformed the UCB-DrAC in eleven games by combining the advantages of multi-view observations and intrinsic reward-driven exploration. UCB-DrAC beat the vanilla PPO agent in ten games and achieved the highest performance in the *Dodgeball* game. In addition, RE3 and RIDE outperformed the vanilla PPO agent in seven games and eleven games, and RIDE achieved the highest performance in the *Ninja* game.

Table 2 and Table 3 illustrate the performance comparison between AIRS and benchmarking schemes on the full Procgen benchmark. For the train levels, the combination of PPO and MEM achieves a higher average episode return than the vanilla PPO agent in most games. Meanwhile, MEM outperformed the UCB-DrAC in eleven games by combining multi-view observations and intrinsic reward-driven exploration advantages. UCB-DrAC beat the vanilla PPO agent in ten games and achieved the highest performance in the *Dodgeball* game. In addition, RE3 and RIDE outperformed the vanilla PPO agent in seven games and eleven games, and RIDE achieved the highest performance in the *Ninja* game.

The game complexity of test levels is much higher than the train levels, which provides a direct measure to evaluate the generalization ability of the agents. PPO+MEM achieved the highest performance in nine out of sixteen games, while UCB-DrAC won three. RE3 obtained the best score in the *Climber* game, while RIDE won the *StarPilot* game. However, the vanilla PPO agent still achieved the highest performance in the *CaveFlyer* game, while the other methods achieved relatively low performance. For a procedurally-generated environment like Procgen games, the agent must quickly adapt to the dynamically changing scenes. To that end, multi-view observations can provide more reference information, while intrinsic rewards allow the agent to comprehend the environment thoroughly. Therefore, MEM can achieve higher generalization ability and facilitate solving real-world problems.

## 6 Conclusion

In this paper, we investigated the visual control problem and proposed a novel method entitled MEM. MEM is the first approach that combines multi-view representation learning and intrinsic reward-driven exploration in RL. MEM first extracts high-quality features from multi-view observations before performing RL on the learned features, which allows the agent to fully comprehend the environment. Moreover, MEM transforms the multi-view features into intrinsic rewards to improve the sample-efficiency and generalization ability of the RL agent, facilitating solving real-world problems with sparse-reward and complex observation space. Finally, we evaluated MEM on various tasks from DeepMind Control Suite and Procgen games. Extensive simulation results demonstrated that MEM could outperform the benchmarking schemes with simple architecture and higher efficiency. This work is expected to stimulate more subsequent research on multi-view reinforcement learning and intrinsic reward-driven exploration.

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

# A    Details on DeepMind Control Suite Experiments

## A.1    Environment Setting

We evaluated the performance of MEM on several tasks from DeepMind Control Suite (Tassa et al., 2018). To construct the multi-view observations, each environment was first rendered using two cameras to form observations of two viewpoints. Then the background of the observation from one viewpoint was removed to form the third viewpoint. Figure 9 illustrates the derived multi-view observations of six control tasks. For each viewpoint, we stacked three consecutive frames as one observation, and these frames were further cropped to the size of $(84, 84)$ to reduce the computational resource request.

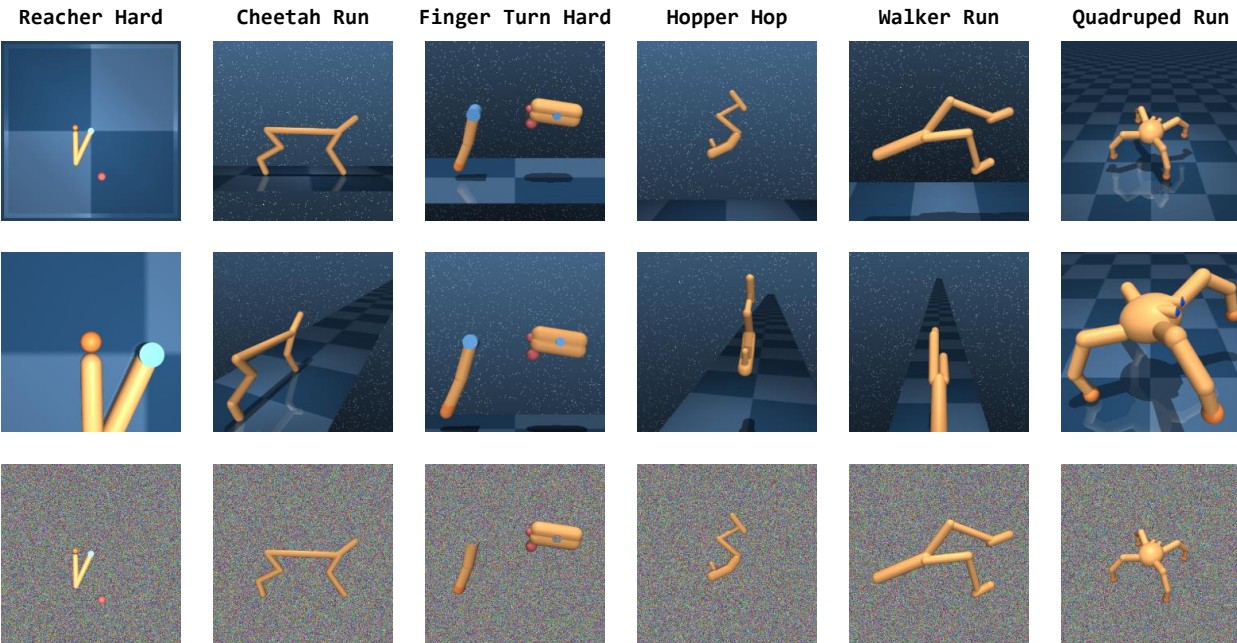

Figure 9: Multi-view observations of DeepMind Control Suite environments.

## A.2    Experiment Setting

**MEM**. In this work, we used the publicly released implementation of SAC (https://github.com/haarnoja/sac) and DrQ-v2 (https://github.com/facebookresearch/drqv2) to update the policy with $r_t^{\text{total}} = \check{r}_t + \beta_t \cdot \hat{r}_t$. For each task, we trained the agent for 500K environment steps and the maximum length of each episode was set as 1000 to compare performance across tasks. Take DrQ-v2 for instance, we first randomly sampled for 2000 steps for initial exploration. After that, we sampled 256 pieces of transitions in each step to compute intrinsic rewards. Then the augmented transitions were used to update the policy using an Adam optimizer with a learning rate of 0.0001 (Kingma & Ba, 2014). As for newly introduced hyperparameters, we used $k = 3$ and performed hyperparameter search over the initial exploration degree $\beta_0 \in \{0.01, 0.05, 0.1\}$, and the decay rate $\kappa \in \{0.00001, 0.000025, 0.00005\}$. We found that the best values were $\beta_0 = 0.05$ and $\kappa = 0.000025$, which were used to obtain the results reported here.

Figure 10 illustrates the employed architectures of the encoding network, policy network, and value network. Four convolutional layers with a ReLU function were used to extract features from the multi-view observations, and two separate linear layers were used to generate the shared features and specific features, respectively. Here the latent dimension $p$ of the shared and specific features was set as 64, and a layer normalization operation was applied to the mini-batch. After that, a linear layer with a softmax function was used to output the classification score and compute the adversarial losses. For the policy network and value network, they accepted the concatenation of the shared and specific features as input and used three linear

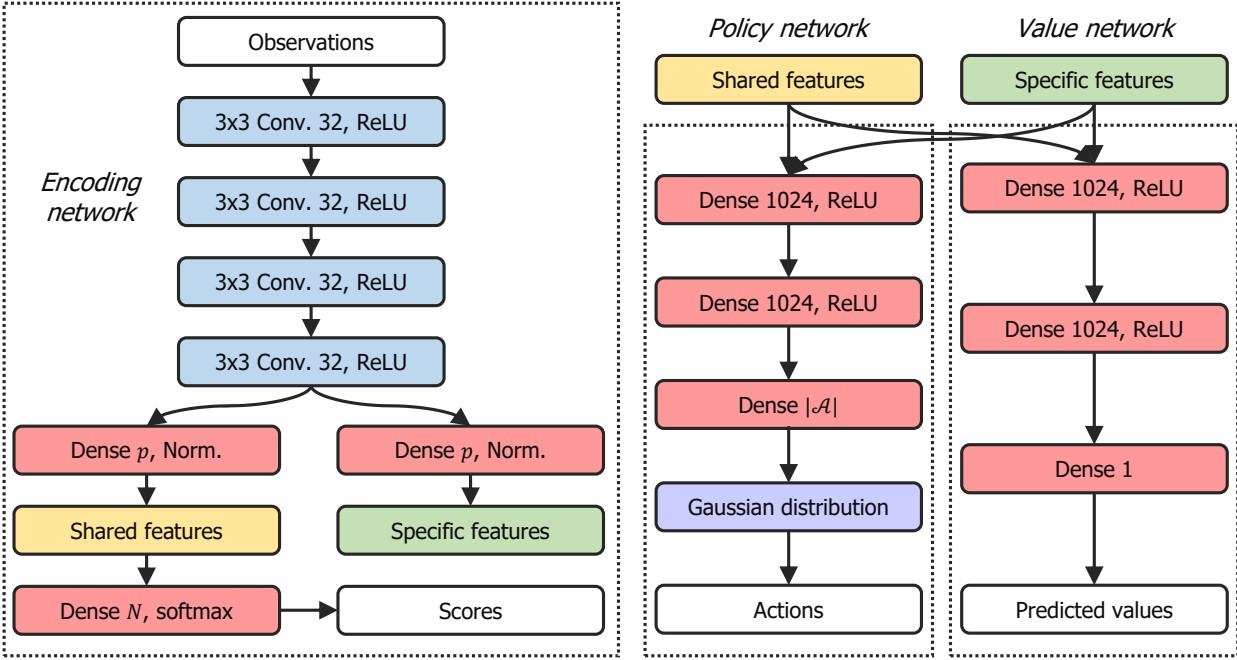

Figure 10: The networks architectures of the multi-view encoder, policy network, and value network.

layers to generate actions and predicted values, respectively. More detailed parameters of DrQ-v2+MEM and SAC+MEM can be found in Table 5 and Table 4.

Table 4: Hyperparameters of DrQ-v2+MEM used for DeepMind Control Suite experiments.

| Hyperparameter | Value |
| --- | --- |
| Number of viewpoints | 3 |
| Gamma | 0.99 |
| Maximum episode length | 1000 |
| Observation downsampling | (84, 84) |
| Stacked frames | No |
| Action repeat | 2 |
| Environment steps | 500000 |
| Replay buffer size | 500000 |
| Exploration steps | 2000 |
| Seed frames | 4000 |
| Optimizer | Adam |
| Batch size | 256 |
| Learning rate | 0.0001 |
| $n$-step returns | 3 |
| Critic Q-function soft-update rate | 0.01 |
| Latent dimension $p$ | 64 |
| $k$ | 3 |
| Margin | 1.0 |
| $\beta_0$ | 0.05 |
| $\kappa$ | 0.000025 |
| Number of eval. episodes | 10 |
| Eval. every steps | 10000 |

Table 5: Hyperparameters of SAC+MEM used for DeepMind Control Suite experiments.

| Hyperparameter | Value |
|---|---|
| Number of viewpoints | 3 |
| Gamma | 0.99 |
| Maximum episode length | 1000 |
| Observation downsampling | (84, 84) |
| Stacked frames | No |
| Action repeat | 4 |
| Environment steps | 500000 |
| Replay buffer size | 100000 |
| Exploration steps | 1000 |
| Optimizer | Adam |
| Batch size | 256 |
| Learning rate of actor & critic | 0.001 |
| Initial temperature | 0.1 |
| Temperature learning rate | 0.0001 |
| Critic Q-function soft-update rate | 0.01 |
| Critic encoder soft-update rate | 0.05 |
| Critic target update frequency | 2 |
| Latent dimension $p$ | 64 |
| Margin | 1.0 |
| $k$ | 3 |
| $\beta_0$ | 0.05 |
| $\kappa$ | 0.000025 |
| Number of eval. episodes | 10 |
| Eval. every steps | 10000 |

**DRIBO**. (Fan & Li, 2022) For DRIBO, we followed the implementation in the publicly released repository (`https://github.com/BU-DEPEND-Lab/DRIBO`). To get multi-view observations, the environment was rendered using the 0-th camera, and a "random crop" operation was applied to the rendered images. During training, the encoder was trained using a combination of DRIBO loss and Kullback–Leibler (KL) balancing, and the weight of KL balancing was slowly increased from 0.0001 to 0.001. At the beginning of training, the agent first randomly sampled for 1000 steps for initial exploration. The replay buffer size was set as 1000000, the batch size was set as $8 \times 32$, and an Adam optimizer with a learning rate of 0.00005 was used to update the policy network. In addition, the initial temperature was 0.1, and the target update weights of Q-network and encoder were 0.01 and 0.05, respectively.

**RE3**. (Seo et al., 2021) For RE3, we followed the implementation in the publicly released repository (`https://github.com/younggyoseo/RE3`). Here, the intrinsic reward is computed as $\hat{r}_t = \|e_t - \tilde{e}_t\|_2$, where $e_t = g(s_t)$ and $g$ is a random and fixed encoder. The total reward of time step $t$ is computed as $r^{\mathrm{total}} = \check{r}_t + \beta_t \cdot \hat{r}_t$, where $\beta_t = \beta_0(1 - \kappa)^t$. As for hyperparameters related to exploration, we used $k = 3$, $\beta_0 = 0.05$ and performed hyperparameter search over $\kappa \in \{0.00001, 0.000025\}$. Finally, the policy was updated using DrQv2.

**CURL**. (Srinivas et al., 2020) For CURL, we followed the implementation in the publicly released repository (`https://github.com/MishaLaskin/curl`). To obtain observations, the environment was rendered using the 0-th camera to generate 100×100 pixel images. To generate the query-key pair for contrastive learning, we used the "random crop" of (Laskin et al., 2020) to perform the image augmentation, which crops the original observations randomly to 84×84 pixels. During training, the replay buffer size was set as 100000, the batch size was 512, the critic target update frequency was 2, and an Adam optimizer with a learning rate of 0.0001 was utilized.

### A.3 Computation Efficiency

We compare the computation efficiency between MEM and benchmarks considering training speed and computational source request. All the experiments were performed using a AMD Ryzen9 7950X CPU and a NVIDIA RTX4090 GPU.

Table 6: Computation efficiency comparison for training 500000 frames.

| Method | Frames Per Second | Training Time | GPU Memory |
|---|---|---|---|
| DrQ-v2 | 135.8 | 5016.9s | 2518MiB |
| DrQ-v2+RE3 | 125.6 | 4063.4s | 2610MiB |
| DrQ-v2+MEM | 75.4 | 15051.2s | 3314MiB |

As shown in Figure 6, MEM does not significantly increase the consumption of GPU memory compared as compared to the vanilla DrQ-v2 agent. But the training speed of MEM is lower than DrQ-v2+RE3 and DrQ-v2. This is mainly because MEM needs to process image inputs at the same time, and multi-view representation learning also produces more training time costs. However, the loss of computational efficiency is worthwhile considering the significant improvement in sampl-efficiency and policy performance. We will further optimize the architecture to improve efficiency while ensuring performance in future work.

# B  Details on Procgen Games Experiments

## B.1  Environment Setting

We evaluated the generalization ability of MEM on various Procgen games (Cobbe et al., 2020). We followed (Raileanu et al., 2021) to construct the multi-view observations using visual augmentations. In particular, the augmentation types were selected based on the best-reported augmentation types for each environment in Raileanu et al. (2021), which is shown in Table 7. Figure 11 illustrates the derived multi-view observations of six Procgen games. For each viewpoint, the frames were cropped to the size of $(64, 64)$ to reduce the computational resource request.

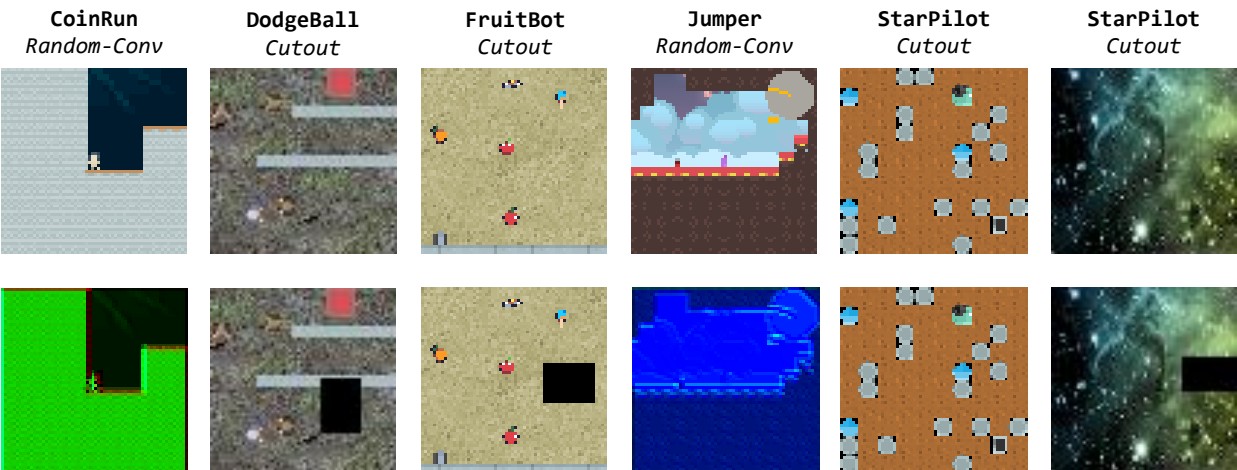

Figure 11: Multi-view observations of DeepMind Control Suite environments.

Table 7: Augmentation type used for each game.

| Game | BigFish | StarPilot | FruitBot | BossFight |
|---|---|---|---|---|
| **Augmentation** | crop | crop | crop | flip |
| **Game** | Jumper | Chaser | Climber | Dodgeball |
| **Augmentation** | random-conv | crop | color-jitter | crop |
| **Game** | Ninja | Plunder | CaveFlyer | CoinRun |
| **Augmentation** | color-jitter | crop | rotate | random-conv |
| **Game** | Heist | Leaper | Maze | Miner |
| **Augmentation** | crop | crop | crop | color-jitter |

## B.2  Experiment Setting

**MEM**. In this work, we used the publicly released implementation of PPO (`https://github.com/ikostrikov/pytorch-a2c-ppo-acktr-gail`) to update the policy with $r_t^{\text{total}} = \check{r}_t + \beta_t \cdot \hat{r}_t$. For each task, we trained the agent for 25 million environment steps with 64 parallel environments. In each episode, the agent first sampled for 256 steps before computing the intrinsic rewards. After that, the augmented transitions were used to update the policy using an Adam optimizer with a learning rate of 0.0005 (Kingma & Ba, 2014). For multi-view encoder training, we used similar network architectures and procedures in Appendix A. As for newly introduced hyperparameters, we used $k = 5$ and performed hyperparameter search over the initial exploration degree $\beta_0 \in \{0.01, 0.05, 0.1\}$, and the decay rate $\kappa \in \{0.00001, 0.000025, 0.00005\}$. We found that the best values were $\beta_0 = 0.1$ and $\kappa = 0.000025$, which were used to obtain the results reported here. More detailed parameters of PPO+MEM are provided in Table 8

Table 8: Hyperparameters of PPO+MEM used for Procgen experiments.

| Hyperparameter | Value |
| --- | --- |
| Number of viewpoints | 2 |
| Observation downsampling | (64, 64) |
| Gamma | 0.99 |
| Number of steps per rollout | 265 |
| Number of epochs per rollout | 3 |
| Number of parallel environments | 64 |
| Stacked frames | No |
| Environment steps | 25000000 |
| Reward normalization | Yes |
| Clip range | 0.2 |
| Entropy bonus | 0.01 |
| Optimizer | Adam |
| Batch size | 256 |
| Learning rate | 0.0005 |
| Latent dim $p$ | 128 |
| Margin | 1.0 |
| $k$ | 5 |
| $\beta_0$ | 0.1 |
| $\kappa$ | 0.00001 |
| Number of eval. episodes | 10 |
| Eval. every steps | 10000 |

**UCB-DrAC**. (Raileanu et al., 2021) For UCB-DrAC, we followed the implementation in the publicly released repository (`https://github.com/rraileanu/auto-drac`). For each update, we first sampled a mini-batch from the replay buffer before selecting a data augmentation method from "crop", "random-conv", "grayscale", "flip", "rotate", "cutout", "cutout-color", and "color-jitter". The exploration coefficient was set as 0.1 and the length of sliding window was set as 10. After that, the augmented observations were sent to compute the data-regularized loss with a weighting coefficient of 0.1. Finally, the policy was updated following the PPO pattern.

**RE3**. (Seo et al., 2021) Here, the intrinsic reward is computed as $\hat{r}_t = \log(\|e_t - \tilde{e}_t\|_2 + 1)$, where $e_t = g(s_t)$ and $g$ is a random and fixed encoder. The total reward of time step $t$ is computed as $r^{\text{total}} = \check{r}_t + \beta_t \cdot \hat{r}_t$, where $\beta_t = \beta_0(1 - \kappa)^t$. Moreover, the average distance of $e_t$ and its $k$-nearest neighbors was used to replace the single $k$ nearest neighbor to provide a less noisy state entropy estimate. As for hyperparameters related to exploration, we used $k = 5$, $\beta_0 = 0.1$ and performed hyperparameter search over $\kappa \in \{0.00001, 0.000025\}$. Finally, the policy was updated using PPO.

**RIDE**. (Raileanu & Rocktäschel, 2020) For RIDE, we followed the implementation in the publicly released repository (`https://github.com/facebookresearch/impact-driven-exploration`). In practice, we trained a single forward dynamics model $g$ to predict the encoded next-state $\phi(s_{t+1})$ based on the current encoded state and action $(\phi(s_t), a_t)$, whose loss function was $\|g(\phi(s_t), a_t) - \phi(s_{t+1})\|_2$. Then the intrinsic reward was computed as $\hat{r}(s_t) = \frac{\|\phi(s_{t+1}) - \phi(s_t)\|_2}{\sqrt{N_{ep}(s_{t+1})}}$, where $N_{ep}$ is the state visitation frequency during the current episode. To estimate the state visitation frequency of $s_{t+1}$, we leveraged a pseudo-count method that approximates the frequency using the distance between $\phi(s_t)$ and its $k$-nearest neighbor within episode (Badia et al., 2020).

## C  MEM with On-policy RL

---

**Algorithm 2:** MEM with On-policy RL

---

Initialize encoding network $F_{\boldsymbol{\theta}}$ and discriminator $D_{\boldsymbol{\phi}}$;

Initialize policy network $\pi_{\boldsymbol{\varphi}}$, maximum number of episodes $E$, coefficient $\beta_0$, decay rate $\kappa$, and replay buffer $\mathcal{B}$;

**for** episode $\ell = 1, \ldots, E$ **do**
 $t \leftarrow 0$;
 **repeat**
  Get multi-view observation $\{\boldsymbol{o}_t^1, \ldots, \boldsymbol{o}_t^N\}$;
  **for** $i = 1, \ldots, N$ **do**
   $\boldsymbol{x}_t^i, \boldsymbol{y}_t^i = F_{\boldsymbol{\theta}}(\boldsymbol{o}_t^i)$;
  **end**
  Get state $\boldsymbol{s}_t = \mathrm{Concatenate}(\boldsymbol{y}_t^1, \ldots, \boldsymbol{y}_t^N, \bar{\boldsymbol{x}}_t)$;
  Sample an action $\boldsymbol{a}_t \sim \pi(\cdot|\boldsymbol{s}_t)$;
  $\mathcal{B} \leftarrow \mathcal{B} \cup \{\boldsymbol{o}_t^{1:N}, \boldsymbol{a}_t, \check{r}_t, \boldsymbol{o}_{t+1}^{1:N}\}$;
  $t \leftarrow t + 1$;
 **until** terminal;
 Update $\beta_\ell = \beta_0(1 - \kappa)^\ell$;
 Get representations $\{\boldsymbol{x}_j^{1:N}, \boldsymbol{y}_j^{1:N}\}_{j=1}^t$;
 **for** $j = 1, \ldots, t$ **do**
  Compute the intrinsic reward $\hat{r}_j$ using Eq. (9);
  Let $r_j^{\mathrm{total}} = \check{r}_j + \beta_\ell \cdot \hat{r}_j$;
 **end**
 Update the policy network with $\{\boldsymbol{o}_j^{1:N}, \boldsymbol{a}_j, r_j^{\mathrm{total}}, \boldsymbol{o}_{j+1}^{1:N}\}_{j=1}^B$ using any on-policy RL algorithms;
 Update $\boldsymbol{\theta}, \boldsymbol{\phi}$ to minimize $L_{\mathrm{total}}$ in Eq. (6).
**end**

---

