# OpenReview forum: "Tackling Visual Control via Multi-View Exploration Maximization"
_TMLR — Rejected by TMLR_

### Review · Reviewer_UL8W · 2022-12-14

**Summary Of Contributions:**

This work proposes a new method to learn of multi-view representations, and also use these representations to do intrinsic reward exploration for RL. Method is tested on multiple benchmarks and compared to a number of other competing methods.

**Audience:**

Yes

**Broader Impact Concerns:**

This particular work mainly focuses on testing in simulated environments and does not have immediate ethical implications.

**Claims And Evidence:**

No

**Requested Changes:**

Major:
- Authors should elaborate on why their results are not consistent with prior works.
- Please explain further why using multi-view representation for intrinsic reward is a better idea than using other exploration method. Preferably with analysis experiments or
- Should add ablation studies and hyperparameter sensitivity studies
- Please add discussion of computation efficiency and whether the added complexity is really worth it

Minor issues:
- page 7 botoom, Finger instead Figer

**Strengths And Weaknesses:**

Strenghts:
- Writing of the paper is mostly clear
- Extensive experiments
- A number of technical details reported
- The paper uses DrQv2, a current SOTA on DMC as a main baseline which is good
- Combining multi-view with intrinsic reward is an interesting idea

Weaknesses:
- **Reliability of results** A major concern is in Figure 3 why your DrQv2 results are very different from those reported by the DrQv2 authors?  For example, in Cheetah run, DrQv2 authors report >800 return after 2M data, in your Figure 3, DrQv2 reaches ~600 at 5M. In Finger turn hard, DrQv2 authors report near 1000 score at 3M, your results reach <200 at 5M. What caused this difference in performance? Same for Figure 5. Authors mentioned they are using images from multiple cameras for DMC, if you are using additional cameras, I believe you should at least get the same performance as DrQv2 since now you have extra information. When you move to this new setting, have you tuned the DrQv2 baseline?
- In Figure 3 Hopper, it seems SAC's performance is much stronger than SAC+MEM? Why is that?
- Proposed method is rather complicated with multiple sophisticated component
- Authors claim learning better representation from multi-view obs can help with harder tasks, which I agree. Have you tested on some of the hardest tasks on DMC? For example, Humanoid tasks?
- R3M has not been tested in Procgen, what are the hyperparameter settings you used for it?
- What about the computation efficiency of your method? The additional complexity and computation can be a concern.
- What is the benefit of using multi-view representation to produce intrinsic reward? Why it will be superior to other intrinsic reward methods?
- Ablation studies and hyperparameter sensitivity studies: what happens when you change one of the hyperparameters? There are a number of critical components, what happens when you shut down one or more of them?

---

> ### Author Response · Authors · 2023-01-24
> **Reply to Reviewer UL8W**
>
> Thank you for taking the time to review our work! We have povided a line by line response below and hope that these can help address the raised comments.
>
> **Q1: Reliability of results**
>
> **A1**:
> Sorry, we wrote the wrong legend in the figure. It should be $10^5$ rather than $10^6$. Since our previous presentation is not very clear, thus we redesigned the experiments and now we have results of six parts based on DrQv2:
> - DrQv2 (singe-view obs)
> - DrQv2 (multi-view obs)
> - DrQv2 (singe-view obs + RE3)
> - DrQv2 (multi-view obs + multi-view representation learning)
> - DrQv2 (multi-view obs + RE3)
> - DrQv2+MEM
>
> Moreover, we included more random seeds and the figures are redrawn. Now the performance comparison is more explicit and reliable. Our MEM produces significant performance gain as compared to the vanilla DrQv2 and RE3 in various tasks. **(See Figure 3, 4, 6, 7, 8)**
>
> **Q2: In Figure 3 Hopper, it seems SAC's performance is much stronger than SAC+MEM? Why is that?**
>
> **A2**: We re-adjusted the hyperparameters and experiments. Now the performance comparison is more accurate.
>
> **Q3: Proposed method is rather complicated with multiple sophisticated component**
> **A3**: Our method has two main components, multi-view encoder and intrinsic reward module. The multi-view encoder only needs a neural network with two branches to output shared and specific information. In addition, the intrinsic reward module directly uses the learned representations of the encoder to compute rewards, which requires no additional models. We provide a computation efficiency comparison in **Appendix C.3**.
>
> **Q4: Authors claim learning better representation from multi-view obs can help with harder tasks, which I agree. Have you tested on some of the hardest tasks on DMC? For example, Humanoid tasks?**
>
> **A4**: Humanoid task takes a lot of time. Due to time constraints, we haven't tried yet and we intend to supplement it in the final version. But we add two more hard robotics manipulation tasks that may benefit from multiple views, namely Reach Site and Place Brick, respectively. Thank you for your understanding. **(See Figure 5 ,6)**
>
> **Q5: R3M has not been tested in Procgen, what are the hyperparameter settings you used for it?**
>
> **A5**: Sorry. What is “R3M”? Now we performed experiments on the full Procgen benchmark **(See Table 2 and Table 3)**.
>
> **Q6: What about the computation efficiency of your method? The additional complexity and computation can be a concern.**
>
> **A6**:  We provide a computation efficiency comparison in **Appendix A.3**.
>
> **Q7: What is the benefit of using multi-view representation to produce intrinsic reward? Why it will be superior to other intrinsic reward methods?**
>
> **A7**: See A1.
>
> **Q8: Ablation studies and hyperparameter sensitivity studies: what happens when you change one of the hyperparameters? There are a number of critical components, what happens when you shut down one or more of them?**
>
> **A8**: For ablation studies, see A1. We performed hyperparameter search over the initial exploration degree $\beta_{0}\in[0.01, 0.05, 0.1]$ and the decay rate $\kappa\in[0.00001,0.000025,0.00005]$. We found that the best values are $\beta_{0}=0.05$, $\kappa=0.000025$, which were used to obtain the latest results reported here.

---

> > ### Comment · Reviewer_UL8W · 2023-02-03
> > **Further comments**
> >
> > Thank you for the rebuttal, the paper has been improved after the rebuttal, and some of my concerns are addressed, however, I have some remaining concerns:
> >
> > - it seems the paper has been through some major changes due to problems in the first version, this kind of major change can make it difficult for reviewers to re-evaluate the paper.
> > - I am not sure I am fully convinced by the rebuttal, for example, the authors did not mention whether they tuned DrQv2 when going into the new setting;
> > - the authors mentioned in A1 they explained why using multi-view representation to produce intrinsic reward can be superior, but the discussion in A1 is very limited, I don't quite see how it answers my question. Additionally this is quite an important question and maybe should be addressed somewhere in the main paper.
> > - lack of results on the hardest tasks is missing which is not ideal (I understand running Humanoid might take longer, but I am not sure it will be that much longer, especially when you are working with visual input...)
> > - The authors mentioned in rebuttal they "re-adjusted the hyperparameters" for some experiments, I am sorry but that is unclear to me how you adjusted them and why that makes the experiments fairer.
> > - Would be good to see some hyperparameter sensitivity study, currently we don't quite know what happens when these parameters are changed, and which ones are the most important if we want to apply this method to a different task.
> >
> > Based on these points that are not fully addressed by the rebuttal, I currently do not feel confident in recommending this paper. However, if the discussion setting is still open, I am happy to further discuss the paper with the authors and other reviewers.

---

### Review · Reviewer_3mwX · 2022-12-23

**Summary Of Contributions:**

The paper proposes a novel multi-view representation learning + exploration algorithm for deep RL. The idea is to collect multiple observations from a partially observable envs (e.g., diff camera views of the same time step) and carry out contrastive learning for representation learning, then on top of the learned representation, an intrinsic reward is computed to drive exploration. The paper describes the algorithm in detail and showcases improvements over baseline methods in a few settings.

**Audience:**

Yes

**Broader Impact Concerns:**

No.

**Claims And Evidence:**

Yes

**Requested Changes:**

I have a few detailed questions re the paper.

=== **General question re multi-view** ===

I am not very familiar with the multi-view POMDP/MDP literature but the paper has provided some necessary background. My questions are: what are the technical difficulties / challenges of multi-view POMDP compared to a single POMDP/MDP with the observation space being the concatenated observation of the multi-view POMDP. If we define a POMDP with observation space $o=(o_1,...o_N)$, it seems that mathematically this problem is equivalent to a multi-view problem with $o_i$ for channel $i$, is that right?

The definition of multi-view POMDP also seems to be quite limited to visual control problems where there are multiple cameras, is it fair to say so? There are not that many other applications that can be cast into multi-view POMDP.

=== **Eqn 2** ===

In Eqn 2, the output of the network $F(o_i)$ is split into $x_i,y_i$, the shared info and channel specific info. How do we carry out such an operation in implementation, do we just split the vector output by $F$ in half?

This architecture is interesting because it encourages the network to extract common info from diff views and also accounts for view-specific features.

=== **Experiments** ===

An overall comment: reading through the experiment results, I would want to have a more clear separation of the benefits that come from representation learning and benefits that come from exploration. It seems to me that this paper aims to propose contributions in these two aspects, but the experiment results do not show as clearly the benefits that attribute to rep learning + intrinsic reward.

Another overall comment is that it feels that in many of the comparisons, the improvements of MEM over baseline methods is not as significant. In Fig 3-4, the variations across seeds are pretty big (e.g. especially the bottom row in Fig 3) and in Fig 6 there is pretty much no clear visual separation between curves at all. Given the relatively limited novelty of the work, it is important to demonstrate clear improvements to make a case for the significance of this paper.

=== **Complex tasks** ===

Fig 3-4 shows complex tasks as benchmarks. I wonder in these tasks if exploration is required at all, since prior work mostly uses such benchmark tasks as testbeds for exploration. In this case, I think the comparison would be more clear if MEM turns of the intrinsic reward, and demonstrates only the benefits of representation learning.

=== **Sparse reward tasks** ===

From what I understand "Cartpole Swingup Sparse" and "Cartpole Balance Sparse" are considered complex tasks which require intrinsic reward to drive exploration in Fig 5. But it seems that methods without explicit exploration mechanism in place can also perform quite well, is that right? In Fig 5, we see that DrQ-v2 achieves improvements and learns quite stably in a number of such tasks. DrQ-v2 was all designed to be purely representation learning based and do not have intrinsic reward in place. Given that they can learn reasonably (albeit sometimes more slowly than the proposed method), it puts into questions whether these tasks are good testbeds for exploration at all?

For example, in Fig 5 if we let the experiments run further for DrQ-v2 for "Cartpole Swingup Sparse", it seems that the learning is at a similar level as MEM, which means that even without intrinsic reward and exploration mechanism in place, DrQ-v2 can still obtain similar level of performance. This makes me wonder whether the difference in performance is not due to fundamental issues of exploration, but rather hyper-parameter differences or the use of different representation learning algorithm. It would be good to clarify this in order to confirm the benefits of exploration.

=== **Hyper-parameters** ===

How do authors choose the intrinsic reward combination coeff $\beta$ and weighting coeff for rep learning loss? These hyper-parameters are argubaly tuned to achieve the performance reported in the current paper, and it is good to showcase the sensitivity of the performance to hyper-parameters, especially given that current results already have some amounts of variance.

**Strengths And Weaknesses:**

=== **Strength** ===

The paper is interesting in that as far as I can see, it provides a novel application of contrastive learning method + a novel way to calculate intrinsic reward based on multi-view observations and learned representations. The resulting aggregate algorithm seems to deliver performance gains in some cases.

=== **Weakness** ===

The novelty & significance of the paper is a bit lacking, since the idea of contrastive representation learning + calculating intrinsic reward based on learned representation has been explored quite a lot in prior literature. The novelty consists purely in combining a few existing methods and delivers a new combined system. This is ok if the empirical results can deliver relatively significant improvements, which also seem a bit lacking in the current stage of the paper, as I will discuss more below.

---

> ### Author Response · Authors · 2023-01-24
> **Reply to Reviewer 3mwX**
>
> Thank you for taking the time to review our work! We have povided a line by line response below and hope that these can help address the raised comments.
>
> **Q1: General question re multi-view**
>
> **A1**: The concept of multi-view was first discussed in a paper “Multi-View Reinforcement Learning” of NeurIPS 2019. Acquiring good-enough policies in the multi-view setting is more complex when compared to standard RL due to the increase in sample complexities needed to reason about varying views. If solved, however, multi-view RL will allow for data-fusion, fault-tolerance to sensor deterioration, and policy generalization across domains. In practice, we do construct the observations as you said. But directly learning from such complex observations is always challenging. Thus we need multi-view representation learning to divide and rule.
>
> **Q2: Eqn 2**
>
> **A2**: Yes, that’s very interesting. You just need to set two output branches in the encoding network, and one for outputting the specific information, and the other one for outputting the share information. After that, the network can be trained via the multi-view loss.
>
> **Q3: Experiments & Complex tasks**
>
> **A3**: We redesigned the experiments and now we have results of six parts based on DrQv2:
> - DrQv2 (singe-view obs)
> - DrQv2 (multi-view obs)
> - DrQv2 (singe-view obs + RE3)
> - DrQv2 (multi-view obs + multi-view representation learning)
> - DrQv2 (multi-view obs + RE3)
> - DrQv2+MEM
> Moreover, we included more random seeds and the figures are redrawn. Now the performance comparison is more explicit and reliable. Our MEM produces significant performance gain as compared to the vanilla DrQv2 and RE3 in various tasks. **(See Figure 3, 4, 6, 7, 8)**
>
> **Q4: Sparse reward tasks**
>
> **A4**: These tasks are relatively simple as compared to Cheetah and Quadruped, thus the benefits of exploration are not very explicit. In contrast, the Procgen games have **procedurally-generated environments**, which have higher requirements for the generalization ability, and the agent has to make sufficient exploration and learn transferable skills rather than memorizing specific trajectories. Therefore, we performed experiments on the full Procgen benchmark **(See Table 2 and Table 3)**.
>
> **Q5: Hyper-parameters**
>
> **A5**: We performed hyperparameter search over the initial exploration degree $\beta_{0}\in[0.01, 0.05, 0.1]$ and the decay rate $\kappa\in[0.00001,0.000025,0.00005]$. We found that the best values are $\beta_{0}=0.05$, $\kappa=0.000025$, which were used to obtain the latest results reported here.

---

### Review · Reviewer_kiU4 · 2023-01-04

**Summary Of Contributions:**

This paper presents a new intrinsic reward approach for deep reinforcement learning (RL). The authors propose to enhance traditional intrinsic reward-driven exploration in RL with multi-view representation learning, which involves representing the environment from multiple viewpoints. The proposed approach uses the estimated entropy across these multiple views as the intrinsic reward to encourage exploration. The authors demonstrate the effectiveness of their approach through experiments on the DeepMind Control Suite and Procgen games, where it outperforms prior intrinsic reward-driven approaches. Overall, the combination of intrinsic reward-driven exploration with multi-view representation learning shows promise as a new approach to deep RL.

**Audience:**

Yes

**Claims And Evidence:**

Yes

**Requested Changes:**

1. Include evaluations on robotic manipulation domains.

2. Run the method with more random seeds and show the statistical significance of the results.

**Strengths And Weaknesses:**

Strengths:

1. The idea of leveraging multi-view representation learning in intrinsic reward-driven RL is interesting and has good impact in the community.

2. The empirical results of the paper clearly show that the proposed method MEM can outperform prior works in many settings especially the DMC tasks.

Weaknesses:

1. I think the improvement of MEM over the prior method RE3 is a bit marginal. In particular, on the Procgen tasks, MEM and RE3 seem to perform almost the same. On DMC tasks, MEM is doing better but I wonder if the authors should include more random seeds as the error bars are quite large.

2. I think the authors should consider evaluating the methods in robotic manipulation tasks where multiple views would be super helpful.

---

> ### Author Response · Authors · 2023-01-24
> **Response to Reviewer kiU4**
>
> Thank you for taking the time to review our work! We have povided a line by line response below and hope that these can help address the raised comments.
>
> **Q1: I think the improvement of MEM over the prior method RE3 is a bit marginal. In particular, on the Procgen tasks, MEM and RE3 seem to perform almost the same. On DMC tasks, MEM is doing better but I wonder if the authors should include more random seeds as the error bars are quite large.**
>
> **A1**: We redesigned the experiments and now we have results of six parts based on DrQv2:
>
> - DrQv2 (singe-view obs)
> - DrQv2 (multi-view obs)
> - DrQv2 (singe-view obs + RE3)
> - DrQv2 (multi-view obs + multi-view representation learning)
> - DrQv2 (multi-view obs + RE3)
> - DrQv2+MEM
>
> Moreover, we included more random seeds and the figures are redrawn. Now the performance comparison is more explicit and reliable. Our MEM produces significant performance gain as compared to RE3 in various tasks. **(See Figure 3, 4, 6, 7, 8)**
>
> **Q2: I think the authors should consider evaluating the methods in robotic manipulation tasks where multiple views would be super helpful.**
>
> **A2**: We very much agree with that because robotic manipulation tasks need more spatial location information to achieve accurate positioning and movement. Therefore, we add two more (We already have Reach Duplo) manipulation tasks of DMC suite, namely Reach Site and Place Brick, respectively. The simulation results demonstrate that multi-view information can significantly improve the policy performance. **(See Figure 5, 6)**
> Due to time constraints, we’d like to test MEM on more manipulation tasks in future work.

---

### Author Response · Authors · 2023-01-24
**Response to all reviewers**

Dear all reviewers and editors,

Thank you for agreeing to postpone the deadline for us! We have carefully discussed your suggestions and revised the manuscript. Hope that these can help address the raised comments.

If you have any further comments, please don't hesitate to let us know. We will try our best to implement it in the final version. Many thanks for taking the time to review our work!

Bests,
Paper642 authors

---

### Decision · Action_Editors · 2023-02-07

**Recommendation:** Reject

**Comment:**

This paper proposes to use a multi-view representation learning architecture for multi-view visual reinforcement learning, with the latent representation being also used for computing an entropy-based intrinsic reward for exploration, this being considered in both off-policy and on-policy actor-critic schemes. This builds heavily on prior literature on contrastive representation learning and intrinsic reward based on learnt representation. The initial reviews raised concerns about technical points and the experiments. The rebuttal addressed some of these concerns, and the revision improved the paper, but not to a sufficient extent to recommend accepting the paper.

Notably, the following points remain, and addressing them could help improving a possible future revision. If DrQv2 is a strong baseline, the paper would benefit from additional baselines based on representation learning or auxiliary losses. Currently, the benefit of exploration is quite questionnable, as there is no striking advantage on sparse reward tasks, and the results on procgen are mixed (when accounting for the std across seeds, notably). The paper could benefit from results on harder tasks, such as humanoid. There are remaining concerns about how baselines are tuned/how fair experiments are (it is not clear how parameters were « re-adjusted »). The advantage of using multi-views could also be further discussed, and the sensitivity to HPs could be further evaluated.

Addressing some further points could also help improving the paper. Some parameters are not discussed at all, notably related to the multi-view part, such as the weighting coefficients of the total loss function or how is trained the min-max loss (gan-like objectives can be hard to train in general). Agents are only trained for 500k steps, while the strongest baseline DrQv2 takes 3M to 5M steps to reach its best performance. It would be helpful to run MEM for a longer time, at least for a subset of tasks. The chosen 3rd view on DM control tasks is not very realistic and seems to provide a lot of information, maybe too much (eg, see «Look where you look! Saliency-guided Q-networks for visual RL tasks. »), it would be good to at least ablate it.

**Audience:**

This submission is relevant to the TMLR's audience.

**Claims And Evidence:**

The claims are not supported enough by accurate, convincing and clear evidence, see detailed comments below.